# Risperidone versus placebo for aggression following traumatic brain injury: a feasibility randomised controlled trial

Shoumitro Deb  ,[1] Lina Aimola,[1] Verity Leeson,[1] Mayur Bodani,[2] Lucia Li,[1] Tim Weaver,[3] David Sharp,[1] Paul Bassett,[4] Mike Crawford[1]

[1]Department of Brain Sciences, Faculty of Medicine, Imperial College London, London, UK
[2]Kent and Medway NHS and Social Care Partnership NHS Trust, Maidstone, UK
[3]Middlesex University, London, UK
[4]Statsconsultancy Ltd, London, UK

**Correspondence to**
Professor Shoumitro Deb;
s.deb@imperial.ac.uk

## ABSTRACT

**Objectives** To conduct a feasibility randomised controlled trial of risperidone for the treatment of aggression in adults with traumatic brain injury (TBI).
**Design** Multicentre, parallel design, placebo controlled (1:1 ratio) double-blind feasibility trial with an embedded process evaluation. No statistical comparison was performed between the two study groups.
**Setting** Four neuropsychiatric and neurology outpatient clinics in London and Kent, UK.
**Participants** Our aim was to recruit 50 patients with TBI over 18 months. Follow-up participants at 12 weeks using a battery of assessment scales to measure changes in aggressive behaviour and irritability (Modified Overt Aggression Scale (MOAS)-primary outcome, Irritability Questionnaire) as well as global functioning (Glasgow Outcome Scale-Extended, Clinical Global impression) and quality of life (EQ-5D-5L, SF-12), mental health (Hospital Anxiety and Depression Scale) and medication adverse effects (Udvalg for Kliniske Undersøgelser).
**Results** Six participants were randomised to the active arm of the trial and eight to the placebo arm over a 10-month period (28% of our target). Two participants withdrew because of adverse events. Twelve out of 14 (85.7%) patients completed a follow-up assessment at 12 weeks. At follow-up, the scores of all outcome measures improved in both groups. Placebo group showed numerically better score change according to the primary outcome MOAS. No severe adverse events were reported. The overall rate of adverse events remained low. Data from the process evaluation suggest that existence of specialised TBI follow-up clinics, availability of a dedicated database of TBI patients' clinical details, simple study procedures and regular support to participants would enhance recruitment and retention in the trial. Feedback from participants showed that once in the study, they did not find the trial procedure onerous.
**Conclusions** It was not feasible to conduct a successful randomised trial of risperidone versus placebo for post-TBI aggression using the methods we deployed in this study. It is not possible to draw any definitive conclusion about risperidone's efficacy from such a small trial.

---

### Strengths and limitations of this study

► We used a stringent randomised controlled trial methodology for a feasibility trial.
► Once recruited in the study, the participants felt supported.
► A process evaluation helped to identify factors that may support or impede successful recruitment in future trials of interventions for post-traumatic brain injury aggression.
► Recruitment has remained a major problem.
► Regulatory authority delays hampered patient recruitment.

---

**Trial registration number** ISRCTN30191436

## INTRODUCTION

A high proportion of patients with traumatic brain injury (TBI) show aggression after the injury (see table 1). Aggression causes distress to patients and their families, can hinder rehabilitation, lead to social isolation, loss of family/friends' support, unnecessary use of restraint and medication, and hospitalisation. In addition, it has financial implications for patients, their families and society as a whole.

The rates of post-TBI aggression vary widely (see table 1) depending on the population studied, definition used for aggression, time after injury, severity of injury and presence of comorbidities. There is no universally accepted definition of aggression, therefore, researchers have used different arbitrary cut-off scores using different rating scales to define aggression which produced a wide range of incidence rates. Also, the boundary between aggression, agitation, irritability, anger and disinhibition is not clear as different studies included different symptoms for assessment. The population point prevalence of post-TBI aggression is unclear. Most studies included patients from clinics or rehabilitation centres, therefore, it is

**Table 1** Rates of post-TBI aggression

| Study | Participants | Follow-up | Assessment | Findings |
|---|---|---|---|---|
| Brooks et al[46] | 134 patients with severe TBI discharged from hospital | 7 years | Researcher designed structured interview | 74% of relatives' reporting and 62% of patients' reporting of anger/irritability/impatience |
| Deb et al[47] | 196 patients with TBI discharged from hospital | 12 months | Researcher designed structured interview | 35.3% patients reported irritability and 15% verbal aggression |
| Tateno et al[4] | 89 TBI vs 26 non-TBI multi-injury patients | 6 months | OAS (researcher determined arbitrary cut-off score) | 33.7% of TBI and 11.5% of non-TBI patients showed aggression. Association with major depression, frontal lobe lesion, poor premorbid social functioning, history of alcohol and substance abuse |
| Dyer et al[3] | 24 TBI vs 21 Spinal Cord Injury (SCI) patients vs 24 uninjured (UI) | Variable | BPAQ | 35%–38% verbal aggression, 25%–39% anger among TBI group. According to informant rating, TBI group showed a higher verbal aggression than SCI and UI group but no intergroup difference in physical aggression |
| Baguley et al[48] | 228 patients with moderate to severe TBI | Variable | OAS | 25% aggressive |
| Draper et al[49] | 53 patients with mild to moderate TBI | 10 years | NFI, FSS, SPRS, HADS | 12% clinically significant aggression score in NFI-aggression. Association with anxiety, depression, fatigue and alcohol use |
| Kelly et al[50] | 190 patients with ABI | Variable | OBS | Verbal aggression: 85.8%; Physical aggression to people: 41.1%; to objects: 35.3%; to self: 5.3% |
| Rao et al[51] | 67 patients with first time TBI | 3 months | OAS | 28.4% aggressive |
| Ciurli et al[52] | 120 patients with severe TBI vs 77 healthy controls | Variable | NPI | 70% verbal aggression and 54% physical aggression to others and objects. TBI group showed a statistically significant higher mean value in NPI-aggression/agitation Scale (p<0.001) |
| Lange et al[53] | Military personnel: Uncomplicated (n=24) and complicated mild TBI (n=17) | 3–6 months | PAI | Aggression in 45.8% in uncomplicated mild TBI and 23.5% complicated mild TBI group |
| James and Young[54] | 152 patients with ABI | Variable | Researcher designed aggression rating scale | 74% verbal aggression and 65% physical aggression |
| Sabaz et al[2] | 507 patients with severe TBI | Variable | OBS | 31.9% aggression |
| Roy et al[6] | 103 patients with first time TBI | 3, 6 and 12 months | OAS | Verbal aggression: 34.3% at 3 months, 41.1% at 6 months and 38% at 12 months. Physical aggression to others: 0% at 3 months, 2.8% at 6 months and 1.4% at 12 months. Aggression to objects: 1.01% at 3 months, 8.2% at 6 months and 7% at 12 months. Self-aggression: 1.02% at 3 months, 5.5% at 6 months and 2.8% at 12 months. Association with depression. |

ABI, acquired brain injury; BPAQ, Buss-Perry Aggression Questionnaire; FSS, Fatigue Severity Scale; HADS, Hospital Anxiety and Depression Scale; NFI, Neurobehavioral Functional Inventory; NPI, Neuropsychiatry Inventory; OAS, Overt Aggression Scale; OBS, Overt Behaviour Scale; PAI, Personality Assessment Inventory; SPRS, Sydney Psychosocial Reintegration Scale; TBI, traumatic brain injury.

difficult to know the true extent of the problem within the TBI population in general as most of them suffer from a mild TBI (70%–85%)[1] and do not attend any clinic or rehabilitation centre. The rate also varies depending on patient or informant rating as informants tend to report a higher rate of aggression than the patients themselves.[2]

Post-TBI aggression is mostly reactive in nature and not premeditated. Verbal aggression and aggression to objects are more common than physical aggression to others.[3]

It is proposed that in the period immediately following TBI, aggression is caused by organic factors such as post-TBI confusion, frontal lobe damage, left cerebral lesion.[4] However, in the chronic stage, psychosocial factors play a more important role in causing aggression.[5 6] Many preinjury and postinjury factors affect the rate and the severity of aggression. For example, premorbid factors such as antisocial personality trait including aggressive tendency, history of psychiatric disorders, impulsivity, alcohol and substance abuse, low educational and socioeconomic status are shown to predispose aggression after TBI.[5 6] Similarly, post-TBI factors such as loss of job, alcohol and substance abuse, comorbid anxiety and depression are shown to be associated with a higher rate of post-TBI aggression.[5 6]

## Neural substrate of aggression

It is postulated that either a reduction in the top-down control from the prefrontal cortex or an overactive limbic system (particularly affecting anterior cingulate gyrus and amygdala) may lead to aggression. These regions are responsible for cognitive interpretation of emotional stimuli which determines a person's response to potentially threatening stimuli. It has been suggested that reduced serotonin and increased dopamine and nor-adrenaline activity in cortical areas, and reduced GABA and acetylcholine activity in limbic areas modulate aggression in humans.[7] Therefore, we can expect that a reduction in cortical dopamine by antipsychotics, nor-adrenaline by beta-blockers and increase in serotonin by selective serotonin reuptake inhibitors, and stabilisation of GABA/glutamate imbalance in the limbic region by antiepileptics should lead to reduced aggression. However, randomised controlled trial (RCT) based evidence does not support these simple assumptions (see table 2). It could be because the role of any particular neuromodulator in aggression is far from clear and they may cause aggression through indirect mechanisms. For example, serotonin is involved in influencing impulsivity and dopamine is responsible for reward punishment behaviour rather than aggression per se, and nor-adrenaline acts as a stress hormone and influences the 'fight-or-flight' response.[8]

## Evidence on pharmacological intervention for post-TBI aggression

Since Deb and Crownshaw's[9] original systematic review on the effectiNeuropsychiatry Inventory-Aggressionveness of psychotropics on neurobehavioural symptoms of TBI,

several reviews have been published on this.[10–19] Findings from all the RCTs on pharmacotherapy for post-TBI aggression are summarised in table 2. A number of studies have shown methylphenidate's (MPH) effectiveness in improving post-TBI cognitive impairment and sometimes apathy symptoms.[14] However, only two very small trials (see table 2) have specifically looked at its effect on post-TBI anger and aggression, one showed superiority of MPH and the other did not.

There are four RCTs involving amantadine, three of which are from the same author (see table 2). Two of the studies used Neuropsychiatric Inventory-Irritability (NPI-I) Scale, which measures irritability rather than aggression.[20] NPI is not an ideal instrument to assess post-TBI aggression as it was originally designed for dementia patients. Also, half of the eight items in NPI-A (NPI-Aggression) subscale do not directly relate to aggression. Most studies showed large placebo effect. Overall finding from these studies is equivocal. One other very small (n=10) cross-over study of amantadine that was reported by a different group did not show any significant intergroup difference in outcome. There is now a major concern about dopaminergic drugs' serious adverse effects such as impulse control disorder like pathological gambling, hypersexuality and binge eating.[21]

There are four very small cross-over trials on beta-blockers, three of which were reported by the same group (see table 2). Three RCTs used cross-over design and one parallel design. Two studies showed superiority of beta-blockers, one was non-significant and one showed partial superiority. These studies have not always used standardised outcome measures and sample sizes are very small. Additionally, both the treatment period and the follow-up are of short duration. Therefore, it is difficult to draw any definite conclusion about beta-blockers' efficacy in treating post-TBI aggression. Furthermore, most studies used a very high dose of beta-blockers which has the potential to cause severe adverse effects such as bronchospasm and hypotension.

## Evidence on psychopharmacological intervention of aggression in non-TBI population

As there is no strong evidence to support any specific pharmacological intervention to treat post-TBI aggression, we extrapolated data from RCTs among other patient groups. Old generation antipsychotics by blocking non-specific dopaminergic activity (particularly involving the nigrostriatal pathway) and also for their anticholinergic effect cause troublesome adverse effects such as extrapyramidal and cardiac symptoms and also show a higher tendency to develop serious adverse effects such as neuroleptic malignant syndrome. On the other hand, new generation antipsychotics provide better tolerability and a broader therapeutic effect by specific action on the D2 dopaminergic system (particularly involving the mesocortical pathway) as well as serotonergic and nor-adrenergic systems. However, they are prone to produce metabolic

**Table 2** RCTs of pharmacological interventions of aggression in post-TBI adults

| Author (year) | Drug dose | RCT design | No of participants | Outcome measure | Findings |
|---|---|---|---|---|---|
| **Psychostimulant** | | | | | |
| Mooney (1993)[55] | Methylphenidate 30 mg daily | Parallel single blind design | Traetment:19 Placebo:19 | POMS-Anger hostility score | Significantly effective in anger reduction |
| Speech (1993)[56] | Methylphenidate 0.3 mg/kg two times a day | Cross-over | 12 closed head injury patients | KAS-belligerence score | NS |
| **Dopaminergic** | | | | | |
| Schnieder (1999)[57] | Amantadine 100–300 mg/day | Cross-over | 10 patients with TBI | Neurobehavioural Rating Scale | NS |
| Hammond (2014)[58] | Amantadine 100 mg two times a day | Parallel design | 76 | NPI-I+NPI-A | NS for the whole group but in a subgroup, amantadine was significantly better |
| Hammond (2015)[59] | Amantadine 100 mg two times a day | Parallel design | 168 | NPI-I | NS, both groups showed large improvements |
| Hammond (2017)[60] | Amantadine 100 mg two times a day | Parallel design | 118 | NPI-I | Two items out of many showed significantly better outcome in the amantadine group |
| **Beta-blockers** | | | | | |
| Greendyke (1986a)[61] | Propranolol 80–520 mg/day | Cross-over | 9 patients with ABI | Observed frequency of aggressive behaviour | Propranolol significantly reduced assaultive behaviour |
| Greendyke (1986b)[62] | Pindolol 10–60 mg/day | Cross-over | 11 patients with ABI (possible overlap with participants in 1986a study; not known) | Observed frequency of aggressive behaviour | Pindolol significantly reduced assaultive behaviour |
| Greendyke (1989)[63] | Pindolol 5–20 mg/day | Cross-over | 10 patients with ABI | OAS | NS |
| Brooke (1992)[64] | Propranolol 60–420 mg/day (recommended maximum is 320 mg/day) | Parallel design | Propranolol:11 Placebo:10 | OAS | Propranolol group showed less intense but same frequency of assaults compared to the placebo group received more physical restraints |

ABI, acquired brain injury; KAS, Katz Adjustment Scale; NPI-A, Neuropsychiatry Inventory-Aggression; NPI-I, NPI-Irritability; NS, non-significant; OAS, Overt Aggression Scale; POMS, Profile of Mood States; RCT, randomised controlled trial; TBI, traumatic brain injury.

syndrome. Therefore, as expected most evidence is based on new generation antipsychotics especially risperidone.

There are six placebo controlled RCTs involving children with autism spectrum disorder and/or intellectual disabilities (ID), all showed superiority of risperidone over placebo (number needed to treat is 3).[22] There are three small RCTs among adults with ID, two showed superiority of risperidone (0.5–4 mg/day) and one showed no significant intergroup difference among risperidone,

haloperidol and placebo.[23] There are two large RCTs both showing superiority of risperidone over placebo in treating aggression in patients with schizophrenia.[24] There are seven RCTs of risperidone (0.5–2 mg/day) showing its superiority over placebo in treating behaviour and psychiatric symptoms of dementia (BPSD), five of them showed specific effect on aggression.[25] However, because of a high rate of cerebrovascular mortality and extrapyramidal adverse effects associated with antipsychotics these drugs

 Deb S, *et al. BMJ Open* 2020;**10**:e036300. doi:10.1136/bmjopen-2019-036300

are not recommended for use for BPSD. Some preliminary evidence also exists in favour of olanzapine, aripiprazole and clozapine in treating aggression in different patient groups which we have not discussed here because of lack of space.[26] At present no strong evidence exists in support of other psychotropics such as mood stabilisers, antidepressants, benzodiazepines.[26 27]

## Justification for our study design

As there is no strong evidence to support any specific drug for the treatment of aggression in TBI, but evidence shows that risperidone may reduce aggression among people with other neuropsychiatric conditions, we set out to examine the feasibility of conducting a placebo controlled trial of risperidone among adults who show aggression following TBI.

## AIM

1. To assess feasibility of a future definitive placebo controlled RCT of risperidone for post-TBI aggression in adults.
2. Perform a sample size calculation needed for a future definitive RCT.
3. Through process evaluation identify factors that may support or impede successful recruitment in future trials.

## METHODS

We conducted a multicentre parallel design placebo controlled (1:1 ratio) double-blind feasibility RCT of risperidone with an embedded process evaluation.

We have summarised the design in this paper. For a full description please see our trial protocol paper.[28] Aggression was defined as any form of behaviour directed towards the goal of harming or injuring another living being who is motivated to avoid such treatment,'[29] and included verbal aggression, and physical aggression to others, property and self, excluding agitation. Rather than using an arbitrary cut-off score on a rating scale, we asked clinicians to refer any patient to the trial that they were considering prescribing medication for the treatment of post-TBI aggression.

## PATIENT AND PUBLIC INVOLVEMENT

The study was developed jointly with a group of 18 adults with TBI from a Headway centre in London, UK. They were not participants of the study. A patient with TBI was part of the project group. An advisory group of patients with TBI who did not take part in the study met every few months to discuss the project and provide feedback through a facilitator (clinical psychologist). They attended a day centre which was not associated to any of the recruiting centres.

## RECRUITMENT

Our aim was to recruit from four centres in London, UK. These included adults with TBI who attended outpatient

clinics, three saw patients with various neuropsychiatric disorders including patients with TBI and one was a neurology clinic dedicated to adult patients with TBI only. The researcher completed a baseline prerandomisation outcome assessment on those patients who agreed to take part. A local clinician completed the Clinical Global Impression-Severity (CGI-S) scale[30] and Udvalg for Kliniske Undersøgelser (UKU) adverse effect scale[31] and arranged for appropriate physical examination. Relevant blood test results were supposed to be collected from participant's case notes if available.

## SAMPLE SIZE

As this was a feasibility study, we did not carry out a formal power calculation but estimated that 40–50 is a reasonable number for a feasibility study which would allow us to carry out a sample size calculation for the full future proposed RCT.[32] A general guide for number of participants necessary for a feasibility study is 30–50 patients.[33]

## Inclusion/exclusion criteria

### Inclusion criteria

1. Aged between 18 and 65 years.
2. Patients with a confirmed clinical diagnosis of TBI using the Mayo Clinic criteria,[34] which happened at least 6 months prior to recruitment.
3. Referred to the clinicians for the management of aggression and for whom the clinician is considering a pharmacological intervention for aggression after investigating and addressing physical, psychological and social triggers.

### Exclusion criteria

1. Patient suffering from post-traumatic amnesia.
2. Comorbid serious mental illness such as schizophrenia and other psychoses, bipolar disorder, major depressive disorder, personality disorder and dementia.
3. Patients who were already on an antipsychotic drug or any other drug that may interact with risperidone at the time of randomisation.
4. Any other contraindication for using risperidone including a previous history of severe adverse events.
5. Patients with no fixed abode or any other reason for which compliance with study medication and monitoring could pose a major problem.
6. A history of definite neurogenic seizures within the last 3 months.

## ACTIVE DRUG DOSE

We used a flexible dose of risperidone. We started with 1 mg once daily dose and increased the dose if necessary, to up to 4 mg/day. Equivalent number of placebo capsules were also administered to the appropriate participants. The placebo capsules looked exactly the same as the risperidone capsules. The researcher followed up participants by telephone every week to assess improvement or

emergence of any adverse events using informally a scale similar to CGI.[30] The researcher liaised with the local clinician every week to provide feedback on the participant's progress. The local clinicians decided whether a dose increase or decrease is warranted based on the information provided by the researcher and where necessary by personally reviewing the participant in their clinic. Local clinicians were allowed to use PRN (as required) medication if necessary. The researcher provided participants with a proforma to keep a log of medication intake. The researcher checked this during the weekly telephone follow-up. At the end of the trial the researcher counted the number of any leftover medicine to assess adherence.

## OUTCOME MEASURES

### Feasibility outcome measures
Our criteria for demonstrating success of the feasibility study were: (1) recruitment of at least 80% of the sample who met inclusion criteria after screening, and (2) completion of follow-up assessments at 12 weeks by at least 75% of the randomised participants.

### Clinical outcome measures

#### Primary outcome measure
We used Modified Overt Aggression Scale (MOAS) to assess improvement in aggression. The MOAS is a simple and widely used 4-item scale that measures verbal aggression along with physical aggression towards other people, property and self. The MOAS[35] is a valid and reliable instrument which is validated in the TBI population[36]

| Table 3 | Patient demographics | | |
|---|---|---|---|
| **Variable** | **Category** | **Placebo** | **Risperidone** |
| No patients | – | 8 | 6 |
| Gender | Female | 3 (38%) | 1 (17%) |
| | Male | 5 (62%) | 5 (83%) |
| Age | – | 43.1±11.3 years | 39.3±8.7 years |
| TBI cause | Assault | 2 (25%) | 0 (0%) |
| | Fall | 3 (38%) | 1 (17%) |
| | Road Traffic Accident | 3 (38%) | 3 (50%) |
| | Other | 0 (0%) | 2 (33%) |
| Initial severity* | Mild | 2 (33%) | 1 (17%) |
| | Moderate | 1 (17%) | 3 (50%) |
| | Severe | 3 (50%) | 2 (33%) |
| History of coma | No | 5 (62%) | 3 (50%) |
| | Yes | 3 (38%) | 3 (50%) |
| History of PTA | No | 5 (62%) | 2 (33%) |
| | Yes | 3 (38%) | 4 (67%) |

Summary statistics are: mean±SD or number (percentage).
*Missing data for two patients in Placebo group.
PTA, post-traumatic amnesia; TBI, traumatic brain injury.

and has been used widely in the assessment of post-TBI aggression and its treatment (see tables 1 and 2). Lower MOAS values indicate lower levels of aggression.

### Secondary outcome measures
1. Glasgow Outcome Scale-Extended version (GOS-E),[37] which is a widely used global outcome measure of post-TBI functioning,
2. Irritability Questionnaire (IRQ),[38] (patient and carer version), this is a well-validated scale and is used in this trial as irritability is often associated with aggression.
3. Hospital Anxiety and Depression Scale (HADS),[39] which is a well standardised easy to use scale for the assessment of depressive and anxiety symptoms. We used HADS as both anxiety and depressive symptoms are common after TBI and we have previously established its cut-off score for brain injury patients.[40]
4. CGI-Improvement (CGI-I) and CGI-S Scales[30] which are well validated and are used widely as secondary outcome measures in clinical trials,
5. UKU Scale[31] to assess adverse effects of risperidone.
6. Two widely used quality of life (QoL) measures, namely EQ-5D-5L[41] and SF-12[42] to assess participant's QoL as the ultimate goal of the intervention is not only symptoms reduction but to improve participant's QoL. These two measures would be useful in calculating quality-adjusted life years in any future definitive RCT.

All these measures were used at baseline and at 12 weeks follow-up.

We used a proforma specifically designed for this study to collect demographic data such as age, gender and also other relevant information such as cause and severity of TBI.

Researcher completed MOAS based on information from both the patients and the carers (where available).

### Concealment and randomisation
Remote web-based randomisation was undertaken through a fully automated service operated by an external organisation 'Sealed Envelope'. Our aim was to randomise an equal number of participants to risperidone and placebo. We used random permuted blocks stratified by study centres (n=4). Sealed Envelope designed the randomisation list and labelled all medicine bottles including placebo with randomisation codes before sending to study centres. There was a 24-hour telephone number available for the clinicians and research team if the randomisation code needed to be broken. Everyone involved in the study including the participants, their carers, local clinicians, researchers, general practitioners were blind to this allocation.

### Statistical analyses of clinical data
The primary outcome was the patient-reported MOAS score, measured at 12 weeks follow-up. As this is a feasibility study, no formal hypothesis tests were performed to statistically compare the two study groups, but descriptive data are presented.

All secondary outcomes were analysed descriptively, with no formal hypothesis tests performed. The frequency of adverse events as per the UKU Scale was calculated at follow-up.

## Process evaluation

A process evaluation using a qualitative method examined trial recruitment, and the feasibility and acceptability of trial procedures from the perspective of patients and their carers. We undertook semistructured qualitative interviews with trial participants, and their carers, purposively sampled to represent site and treatment allocation.

Interviews were conducted using a bespoke topic guide. As the qualitative interviews generated a rich dataset it is beyond the scope of this paper to present all the details here. The topic guide was developed under the following broad headings; (1) Background information (how TBI has affected participants' lives in general), (2) Interventions (was it difficult to comply with the intervention?), (3) Outcome (how did they feel about the overall outcome measures at the end of the study?), (4) Trial procedures (taking part in questionnaire completion, support from the research team etc), (5) Factors influencing participation and retention (eg, factors hampering vs helping with

participation). These interviews were audio-recorded, transcribed and analysed using a thematic framework approach[43] and managed using NVivo[44] computer software.

## RESULTS

The demographic characteristics of study participants are summarised in table 3. The mean age of patients was 43 and 39 in the placebo and risperidone groups respectively. Over 60% of both groups were male. Fourteen of the 43 (32.6%) eligible patients were recruited between March 2017 and January 2018 (see figure 1). Eight patients received placebo and six risperidone (see figure 1). However, two participants from the placebo group and one from the risperidone group withdrew. Two withdrew due to adverse events and for one participant follow-up data were available. Therefore, MOAS data were calculated for five risperidone and seven placebo group participants (see table 4), giving an 85.7% follow-up rate. Verbal aggression was the most common type of aggression exhibited by the participants along with aggression to property, but a minority also showed aggression directed

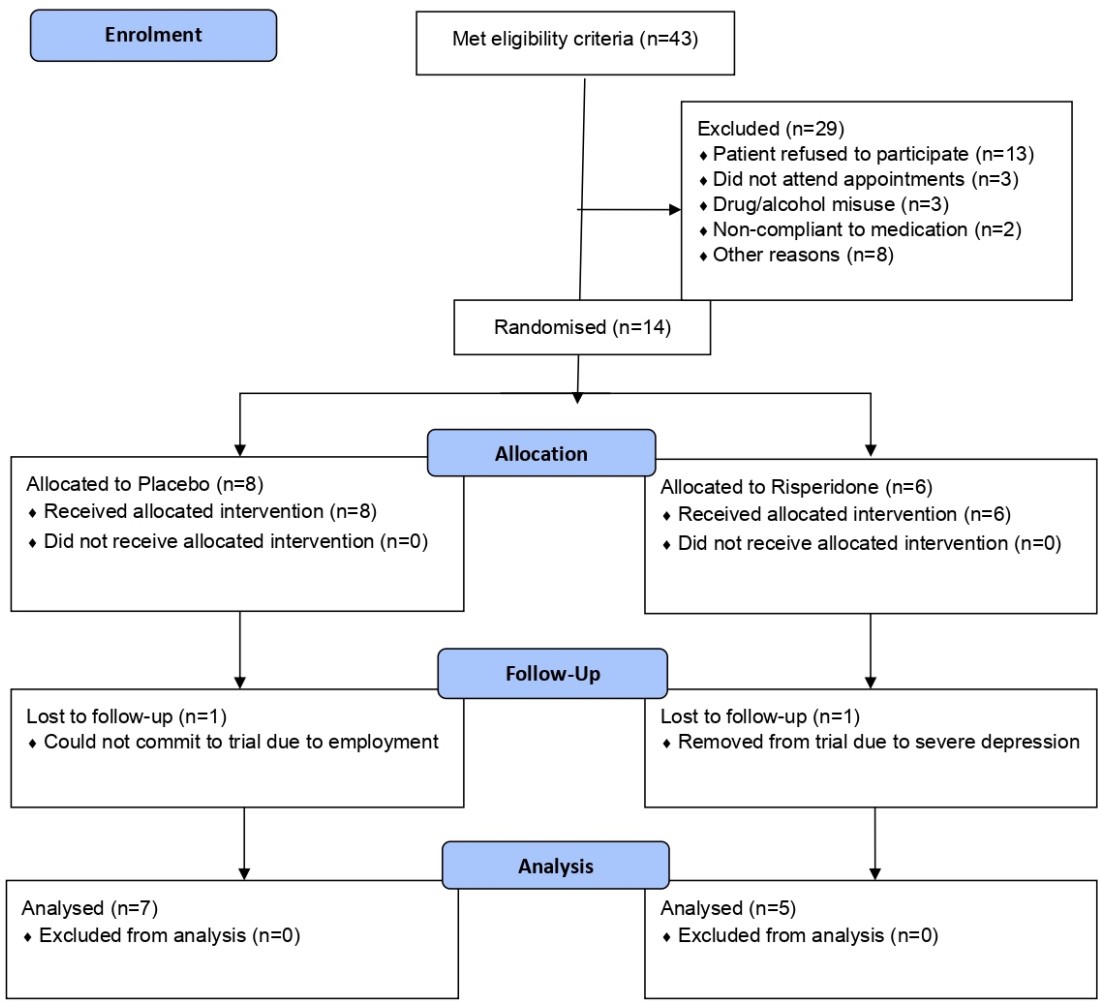

**Figure 1** Flow chart of study recruitment.

**Table 4** Primary outcome MOAS patient ratings

| Time point | Group | N | Mean±SD | Median (IQR) | Range |
|---|---|---|---|---|---|
| Baseline* | Placebo | 7 | 11.6±9.6 | 8 (2 to 24) | 2 to 25 |
| | Risperidone | 5 | 5.0±3.0 | 6 (2 to 8) | 2 to 9 |
| Week 12 | Placebo | 7 | 2.0±2.4 | 1 (0 to 3) | 0 to 7 |
| | Risperidone | 5 | 2.2±3.9 | 0 (0 to 6) | 0 to 9 |
| Change† | Placebo | 7 | −9.5±10.5 | −4 (−23 to 1) | −25 to 1 |
| | Risperidone | 5 | −2.8±4.8 | −2 (−8 to 2) | −9 to 3 |

*Summary statistics only for patients with values at 12 weeks.
†Calculated as value at week 12 minus value at baseline.
MOAS, Modified Overt Aggression Scale.

towards self and others. Three participants received 2 mg and two received 4 mg risperidone at the end point. No postintervention blood test results were available from participants' case notes. No one had a psychiatric diagnosis such as schizophrenia, major depressive disorders, anxiety disorders, personality disorder or dementia. No one received concomitant psychotropic medication or PRN (as required) medication.

### Feasibility outcome data
Our original recruitment target was 50 participants from four centres over 18 months (0.69 patient/centre/month). One centre did not recruit any patient and another centre recruited only one patient who did not complete the trial. The overall recruitment rate was 0.3 patient/centre/month across all four centres (21% of the target rate) over 10 months period. The recruitment rate

from a research active University centre was 0.78/month (113% of the target rate). This centre was dedicated to adults with TBI and held a patient database which helped the researcher to screen for potential recruits. Because of regulatory authority delays the total recruitment time available was shortened by 8 months.

### Quantitative outcome data
The data suggest slightly higher MOAS values at baseline in the placebo group. At 12 weeks, MOAS values had reduced in both groups, to a similar level (see tables 4 and 5). Therefore, the placebo group showed a larger change in total MOAS values and its subscores. In order to aid a possible future sample size calculation (using an analysis of covariance approach), the association between MOAS total score at baseline and 12 weeks was assessed. Spearman's rank correlation gave a correlation of 0.04 between

**Table 5** MOAS subdomain scores

| Outcome | Time point | Placebo N | Placebo Summary | Risperidone N | Risperidone Summary |
|---|---|---|---|---|---|
| MOAS patient rating | Baseline* | 7 | 4.7±2.4 | 5 | 3.4±1.7 |
| Verbal | Week 12 | 7 | 1.4±1.3 | 5 | 1.0±1.4 |
| | Change† | 7 | −3.3±2.7 | 5 | −2.4±2.3 |
| MOAS patient rating | Baseline* | 7 | 2.0±2.1 | 5 | 0.2±0.4 |
| Against property | Week 12 | 7 | 0.3±0.8 | 5 | 0.2±0.4 |
| | Change† | 7 | −1.7±2.4 | 5 | 0.0±0.0 |
| MOAS patient rating | Baseline* | 7 | 0.6±0.8 | 5 | 0.4±0.9 |
| Autoaggression | Week 12 | 7 | 0.0±0.0 | 5 | 0.0±0.0 |
| | Change† | 7 | −0.6±0.8 | 5 | −0.4±0.9 |
| MOAS patient rating | Baseline* | 7 | 0.3±0.8 | 5 | 0.0±0.0 |
| Against people | Week 12 | 7 | 0.0±0.0 | 5 | 0.2±0.4 |
| | Change† | 7 | −0.3±0.8 | 5 | 0.2±0.4 |
| MOAS carer rating | Baseline* | 5 | 8.4±8.4 | 3 | 10.3±6.5 |
| Total score | Week 12 | 5 | 2.8±3.1 | 3 | 3.7±3.8 |
| | Change† | 5 | −5.6±5.6 | 3 | −6.7±3.2 |

*Summary statistics only for patients with values at 12 weeks.
†Calculated as value at week 12 minus value at baseline.
MOAS, Modified Overt Aggression Scale.

**Table 6** Secondary outcome measures (IRQ, GOS-E, EQ-5D-5L, SF-12, HADS)

| Outcome | Time point | Placebo N mean±SD | Risperidone N mean±SD |
|---|---|---|---|
| IRQ patient Severity | Baseline* | 7 34±6 | 5 37±19 |
| | Week 12 | 7 26±7 | 5 25±14 |
| | Change† | 7 −8±9 | 5 −13±13 |
| IRQ carer Severity | Baseline* | 5 10±4 | 5 14±5 |
| | Week 12 | 5 7±4 | 5 7±4 |
| | Change† | 3 −3±2 | 5 −6±2 |
| GOS-E | Baseline* | 7 6.4±1.1 | 5 6.6±1.5 |
| | Week 12 | 7 6.4±1.1 | 5 6.8±1.6 |
| | Change† | 7 0.0±0.0 | 5 0.2±0.5 |
| EQ VAS | Baseline* | 7 53±24 | 5 61±10 |
| | Week 12 | 7 57±30 | 5 64±21 |
| | Change† | 7 4±11 | 5 3±18 |
| SF-12 Physical | Baseline* | 7 44±13 | 5 51±10 |
| | Week 12 | 7 43±11 | 5 47±12 |
| | Change† | 7 0±14 | 5 −3±5 |
| SF-12 Mental | Baseline* | 7 34±14 | 5 30±12 |
| | Week 12 | 7 41±15 | 5 40±19 |
| | Change† | 7 7±18 | 5 9±13 |
| HADS anxiety | Baseline* | 7 8.6±3.8 | 5 9.6±6.9 |
| | Week 12 | 7 6.7±3.8 | 5 6.8±7.2 |
| | Change† | 7 −1.9±3.5 | 5 −2.8±1.8 |
| HADS depression | Baseline* | 7 9.9±5.5 | 5 4.8±4.2 |
| | Week 12 | 7 7.9±4.7 | 5 5.0±3.9 |
| | Change† | 7 −2.0±2.8 | 5 0.2±2.9 |

*Summary statistics only for patients with values at 12 weeks.
†Calculated as value at week 12 minus value at baseline.
GOS-E, Glasgow Outcome Scale-Extended; HADS, Hospital Anxiety and Depression Scale; IRQ, Irritability Questionnaire.

timepoints. This suggests little relationship between the scores at the two time points.

The secondary outcome measures showed mixed results (see table 6). Numerically the score changes in the IRQ and HADS-Anxiety were slightly greater in the risperidone group whereas HADS-Depression score change was slightly greater in the placebo group. A similar picture was revealed for the QoL scales scores (SF-12 and EQ-5D-5L), and also GOS-E (see table 6). According to CGI-I, 40% in the risperidone group and 33% in the placebo group, respectively, were 'much' or 'very much improved' at follow-up. Patient MOAS score correlated with both baseline (r: 0.78; p<0.001) and follow-up patient IRQ severity score (r: 0.64; p=0.03).

No severe adverse events were reported. The overall rate of adverse events remained low although it was slightly higher numerically in the risperidone group.

An approximate sample size calculation done by the study statistician (PB) shows that for a 5% significance level and 90% power, 72 patients per group and 144 in total will be required. Allowing for a 20% attrition rate, a total of 180 patients would need to be recruited.

### Qualitative outcome data

Eight patients and two of their carers were interviewed and the data were analysed by the researcher (LA), under the supervision from the qualitative investigator (TW).

Key factors exerting a positive influence on participation reported by patients and their carers included a belief in the positive benefit of research. This may have been expressed as (1) a perception that the trial may help to achieve their 'desire to get better', or (2) satisfy an altruistic motivation to participate in research that might help other people in a similar situation. Some also reported (3) previous positive experiences in taking part in research. Also, significant to maintaining engagement were (4) the straightforward research procedures and (5) the high level of trust and support engendered by weekly telephone contact from the researcher (LA) and (6) good relations with the referring doctor. Participation was also associated with (7) a higher educational level of the participants.

The analysis showed that the key reasons for patients not taking part in the study were: (1) a reluctance to change treatment, (2) perceived adverse effects of the study medication, (3) lack of understanding of the reason why the study is undertaken as this was not clearly explained to them, (4) lack of insight into their own condition, so the potential participants did not see their behaviour as a problem needing any intervention, (5) unwillingness to engage with treatment regime or trial procedures and (6) objection to randomisation manifested as a reluctance particularly on the part of the carers regarding potential allocation into the placebo arm which they thought might make the patient's behaviour worse and difficult to manage.

Furthermore, many participants, particularly those who had memory and attention problem, found it tiring to complete so many assessment measures some of which were quite long. Similarly, memory problems may have affected the consent process thus making some potential patients ineligible for recruitment.

### DISCUSSION

Given the very small sample size, it is not possible to draw any definite conclusion from the interpretation of the outcome data.

### Clinical findings

In the current study, risperidone did not show superiority over placebo at follow-up as measured using total MOAS score (the primary outcome). No clear pattern emerged in secondary outcomes. Like previous studies most of the aggression in our study was primarily verbal in nature.[3] Also, as expected, there was a correlation between patient reported MOAS and IRQ scores suggesting a clinical

association between irritability and aggression. Carer reported mean MOAS values are numerically higher than the patient reported values both at baseline and at follow-up. This is in keeping with previous studies[3] but difficult to comment on given the small sample size and skewed distribution of MOAS values. This could be due to patient's poor insight to their problems, using denial as a psychological defence and impaired self-awareness.[3]

### Risperidone dosage

There was no evidence of non-adherence to medication in this trial. It is difficult to determine the right therapeutic window for the risperidone dosage as a lower dose may be ineffective and a higher dose may produce unwanted adverse effects. This is even more difficult to decide for patients with TBI as there is no research-based evidence to calibrate the dose specifically for this patient group. Therefore, we had to look into dosage used in RCTs for non-TBI patient groups for a reference point. We used a lower dose of risperidone than that used for the treatment of schizophrenia for the following reasons: (1) a high dose of risperidone is likely to lower seizure threshold and the patients with TBI are vulnerable to develop seizures, we, therefore, started low and went slow, (2) reports from one laboratory have shown detrimental effect on experimentally induced TBI rodents' behaviour and cognition from chronic use of high dose risperidone, although this finding has not been replicated in humans[45] (3) RCTs of risperidone among adults with ID[23] and also dementia[25] used a lower dose (2–4 mg/day), (4) in the UK clinicians use low-dose risperidone to treat post-TBI aggression (0.5–4 mg/day) and (5) we were advised by the patient advisory group to use low dose as they were worried about risperidone's adverse effects at a high dose. Also, it appears that a low dose (2 mg/day) was sufficient for most participants in our study.

### Feasibility outcome

The purpose of the study was to assess feasibility of a future definitive RCT. We concluded that it may not be feasible to conduct a definitive trial using the approach we took in this feasibility study. Our rate of recruitment fell short of our original goal because two of the four centres failed to recruit participants. However, if the rate of recruitment achieved in the two active sites, were replicated across other centres, a definitive RCT would be feasible.

We also concluded that an adequate number of participants could not be recruited in a future definitive full RCT from neuropsychiatry clinics alone. We believe that to recruit an adequate number of participants in the future, it would be necessary to broaden the range of services to include neurorehabilitation centres and possibly trauma centres.

Our experience shows that research active centres are likely to recruit an adequate number of participants in a future RCT as they already have the infrastructure and the patients attending these centres are already familiar with research procedures and may have already taken part in other research studies. However, the downside may be that patients may have already been recruited in one or more studies, therefore, would be difficult to recruit into yet another study.

We found the way the project is presented to the potential participant is of utmost importance. Therefore, we feel that a face-to-face meeting with the potential patients explaining the study properly than a telephone call is more likely to increase the chance of recruitment. Regular support such as weekly telephone calls (as was the case in our study) is likely to retain the patient in the study.

We found that it is important to carefully choose exclusion criteria as in our study initial criterion of excluding patients with a history of epilepsy meant exclusion of some otherwise eligible patients. We later revised the exclusion criterion to 'history of a definite neurogenic seizure within the last 3 months.'

### Recommendations from the feasibility study

A successful definitive RCT would require: (1) recruitment through specialised TBI clinics, (2) clinics having databases of potentially eligible patients, (3) a senior clinician engaged enough to facilitate the study and (4) clinical colleagues who are enthusiastic and prepared to help with the study, (5) regulatory approvals in place well before the planned start of recruitment (6) availability of sufficient resources to be able to spend time with potential recruits in a face to face meeting to describe the study, (7) regular support through either regular visits or telephone checks, (8) simple study procedures to optimise retention in the study, for example, make the study medication easily accessible and keep the number of questionnaires and tests to absolute minimum (9) information in an accessible format as many patients with TBI would have difficulty in concentrating and memorising complex information and (10) some form of non-pharmacological intervention such as 'anger management' alongside study medication to increase the recruitment and retention rate.

### Strengths and limitations

The RCT design in our study is robust and would create minimum selection, performance, detection, attrition, reporting, concealment and other biases. Once recruited in the study, the participants felt supported. A process evaluation helped to get positive and negative feedbacks from the participants. However, recruitment remained a major problem and delays in obtaining regulatory approvals hampered patient recruitment. The total number of participants was too small. Although we achieved our feasibility target of 75% retention, we failed to recruit at least 80% of the sample who met inclusion criteria after screening.

**Acknowledgements** The Imperial Biomedical Research Centre Facility which is funded by the National Institute of Health Research, UK provided support for the study. We thank Dr Niruj Agrawal for recruiting participants and participating patients and their carers, and also Clinical Services Officers who helped to collect data. Patient advisory group and their facilitator, Anita Rose.

**Contributors** SD was involved in the conception and design of the study, supervised data collection, and wrote the funding application, the study protocol

and this manuscript. LA contributed to study protocol, collected all research data and liaised with recruiting sites, carried out qualitative interviews with patients and their carers, and contributed to this manuscript. VL contributed to study protocol, overseen regulatory procedures including Ethics and MHRA approval and supervised the day-to-day running of the trial, and contributed to this manuscript. MB contributed to the study protocol, recruited patients and contributed to this manuscript. LL recruited patients and contributed to this manuscript. TW contributed to the study protocol, supervised qualitative interview data analysis, and contributed to this manuscript. DS contributed to study protocol, recruited patients and contributed to this manuscript. PB helped with the study design, developed a statistical analysis plan (SAP) and carried out the statistical analyses, and contributed to this manuscript. MC contributed to the study protocol, supervised day-to-day dealings with the Sponsor and contributed to this manuscript.

**Funding** This article presents independent research funded by the National Institute for Health Research (NIHR), UK under its Research for Patient Benefit (RfPB) Programme (Grant Reference Number PB-PG-1013–32054).

**Disclaimer** The views expressed are those of the authors and not necessarily those of the NHS, the NIHR or the Department of Health, UK.

**Competing interests** None declared.

**Patient consent for publication** Not required.

**Ethics approval** The study received ethical approval from the London-Westminster Research Ethics Committee on behalf of the UK NHS Health Research Authority.

**Provenance and peer review** Not commissioned; externally peer reviewed.

**Data availability statement** Data are available on reasonable request. Only the designated trial investigators have access to the personal data of the participants and to the final dataset. The original e-CRF pages generated during the study is the property of the sponsor (Central and North West London Partnership NHS Foundation Trust, UK). Regulatory bodies if appropriate may request access to personal data. There may be opportunity to share data anonymously if necessary and appropriate after proper authorisation and approval are obtained.

**ORCID iD**
Shoumitro Deb http://orcid.org/0000-0002-1300-8103

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
