## [Reviewer comments · BMJ Open]

ARTICLE DETAILS

TITLE (PROVISIONAL)	Risperidone versus placebo for aggression following traumatic brain injury: a feasibility randomised controlled trial.
AUTHORS	Deb, Shoumitro (Shoumi); Aimola, Lina; Leeson, Verity; Bodani, Mayur; Li, Lucia; Weaver, Tim; Sharp, David; Bassett, Paul; Crawford, Mike

VERSION 1 - REVIEW

REVIEWER	Malcolm Hopwood Department of Psychiatry University of Melbourne Australia
REVIEW RETURNED	28-Jan-2020

GENERAL COMMENTS	This submission describes a well structured and well described RCT of risperidone versus placebo for post ABI aggression. This type of study is clearly needed in this relatively low evidenced area of practice and where the balance of harms and benefits needs careful consideration. The background to the study contains much of the relevant literature. The study design is well laid out and not really open to much criticism. There are several instruments available for the measurement of aggression in this setting and the choice of the MOAS, whilst not unreasonable, could be discussed a little. The dose range utilised in the study is consistent with common clinical practice although it may be worth considering and discussing whether an even lower dose such as 0.5 mg was worth initial trial given the sensitivity issues in this group. The results of the study are laid out well with the main problem in their interpretation being the obvious difficulties with recruitment. The authors are appropriately statistically conservative given the low numbers but do draw some tentative conclusions about the relative efficacy of placebo and risperidone. Interestingly, the results in the risperidone arm were actually numerically better when looking at carer ratings of aggression, and this result should be considered a little more in my view given that we know there is a high level of disparity between patient and carer ratings of aggression in this setting. The major focus of the paper then is on the struggle with recruitment and the discussion around this area is of value as this is a problem that has dogged many studies in this specific area. This may be the most valuable contribution of the paper and is thus definitely worth
---

	retaining.
--	------------

REVIEWER	Durga Roy, MD Johns Hopkins University School of Medicine Department of Psychiatry and Behavioral Sciences
REVIEW RETURNED	30-Jan-2020

GENERAL COMMENTS	While the study question and intervention is interesting, the manuscript is poorly written with a substantial number of grammatical and syntax errors, cursory literature review on TBI aggression and unclear rationale described in reference to parts of the study protocol. The methods appear to be sound, but are not organized with appropriate syllogism to guide the reader as to the rationale for many components of the study design. The most overarching concern is that while this is a feasibility study, the authors use this as a reason for not having completing a power analysis or even establishing a hypothesis. Also there is no discussion of adjusting for confounding factors such as psychiatric comorbidities (not symptoms , but syndromes) that exist with TBI aggression. For a data driven paper, in the very least the reader would most certainly benefit from hearing about what the authors hypothesize would be the outcome of using an intervention like Risperidone to manage aggression. I have listed my comments and questions based on each section: Introduction  - The background literature in the introduction is extremely underdeveloped. Some of the most landmark studies in TBI aggression (Tateno et al, etc) are not cited. Would suggest a more robust lit review on this topic. - In the first paragraph would clarify what is meant by “medical advances” line 45 page 3. - On page 4 line 3, the authors mention that aggression is the most challenging post-TBI neurobehavioral symptom but what is the prevalence? - Aggression is not defined anywhere in the intro. It should also be made very clear that aggression is not the same as agitation and though neuroleptics have been used in the management of both it needs to be made clear that this paper will focus on aggression only. - Line 10 the authors mention “Both pharmacological and non-pharmacological interventions are used to manage this difficult behavior” but there is no reference cited following this sentence to substantiate the claim. This is a key piece of background information as the paper hinges on a pharmacologic intervention to manage TBI aggression. - Though it is clear that there is no high quality evidence of trials, namely they were not RCT, what were the limitations the findings in the literature ? were they only case reports or very underpowered open label studies? - Would explain to the reader why second generation antipsychotics are used to manage aggression in this patient population. – namely the side effect profile - On page 5 line 37- why is it that clinicians tend to choose antipsychotics? Would talk about the literature behind these practices. - Page 5 Line 40 is a run on sentence and needs to be revised, - The rationale behind why the authors chose risperidone as the intervention for this study is not clear. Just because it has been used
--

in the other neurobehavioral syndromes does not necessarily make it an obvious intervention in TBI. It is well documented that higher potency neuroleptics can impair neuronal growth. The authors should delve more into the pharmacokinetic, pharmacodynamic, efficacy and effectiveness reasons that this drug was chosen.

Methods

- In the methods section, subjects were recruited from a Headway center. Is this a residential center? Might be helpful if the setting is clear to the reader whether this is an inpatient, outpatient, rehab or residential setting.
- Line 31 on page 6 does not make grammatical sense
- What is a trials management group and what did it mean that the subject was a co-applicant.
- What was the study feedback from participants done through focus groups? (page 4 line 42)
- I am not clear as to why there was no power calculation. The authors cite the reason as this being a feasibility study, however this should not preclude to the study design to include a power analysis.
- Why were patients with PTA excluded? If there was a need to exclude these patients then what about those that had LOC or AMS? A rationale as to why those with PT were excluded is something that needs to be clarified.
- How was dosing of Risperdal determined? It is reported by authors that dosing was "flexible" but
- Similar to the above comment on no power analysis- in the statistical analysis why was there no hypothesis tested? Page 7 line 36. Even as a feasibility study, it would be useful to the reader if it could be determined what outcomes the authors would expect from a study like this. Which would also require a more thorough review of the literature about what has been done in pharmacological management of TBI aggression.
- On page 8 line 34, what is meant by interviews were conducted by interview topic? This is very ambiguous and requires much more detail. It is mentioned above that a semi-structured

Results

- Why were 80% of patients on risperidone (should not read risperidone patients) mildly or less ill at the 12 week timepoint? Is it because they had a full trial duration of the neuroleptic? Also, what is meant by "mildly ill?" Are these in reference to psychiatric symptoms?
- How was aggression due to psychiatric comorbidity adjusted for? It is well established that aggression can co-occur with depression, mania, anxiety and psychosis in TBI, so how was this incorporated into the analysis.
- Was the placebo group on any other psychotropic? I am not clear about how the intervention was randomized to each group if the goal was to see if aggression was reduced in the risperidone group. There should also be mention of IRB approval somewhere in the text for the study protocol.

Discussion

- This section is quite superficial and doesn't delve into the implications of the study findings. What can the reader/clinician take away from these outcomes? Understood that it is a feasibility study but some could some preliminary conclusions be drawn?
- There are no limitations listed in the discussion and this is a standard component of this section

Overall this paper could be of some value in the first steps of learning about management of TBI aggression but the lack of solid

	scientific rigor in the study design and statistical analyses, very cursory review of the literature on TBI aggression and underdeveloped syllogistic writing style are detrimental to the final product.
--	---

REVIEWER	Professor Jennie Ponsford Monash University, Australia
REVIEW RETURNED	02-Mar-2020

GENERAL COMMENTS	This study promised to fill a significant gap in evidence for pharmacological management of aggression after traumatic brain injury. Unfortunately the study met with many hurdles regarding feasibility. The following comments are intended to improve the manuscript. Abstract:  - Page 2, line 5: It is misleading to say that overall rate of adverse events was non-significantly higher in the risperidone group, as no significance testing was conducted. This might also lead the reader of the abstract to believe that other findings were hypothesis tested, which they weren't. It should also be stated in the abstract that two people withdrew from the study due to adverse events. - Page 2, Line 12: More detail could be added regarding what measures may increase recruitment, even if only a couple. This is very vague at present. - Page 12, 36: It is stated that many regulatory authority delays hampered patient recruitment in the abstract but don't discuss this again. This should surely be added to the discussion if it is significant enough to appear in the abstract. Introduction:  - The published protocol introduction has been copied almost exactly. Published protocol link: https://trialsjournal.biomedcentral.com/articles/10.1186/s13063-018-2601-z - P3 Line 16 to 20 – there are existing published studies examining medication prescribing for neurobehavioural problems post ABI – it would be best to cite one of these rather than unpublished data - P3 Line 23 – A more recent systematic review confirming the lack of high-quality clinical trials in this area is : Hicks, A.J., Clay, F.J., Hopwood, M., James, A., Jayaram, M., Perry, L.A., Batty, R. & Ponsford, J.L. (2019). The Efficacy and Harms of Pharmacological Interventions for Aggression after Traumatic Brain Injury – Systematic Review. Frontiers in Neurology. Published online 29 November 2019. DOI: 10.3389/fneur.2019.01169. - P3 Line 32 – Please justify why second generation antipsychotics are chosen more frequently than first generation compounds. - P3 Line 39 –Please include a reference for the claim that clinicians use their 'often limited experience' to choose antipsychotics to treat aggression post TBI - P3 Line 45 (and elsewhere) – Further explanation is required to justify the decision to administer low dose risperidone. It was indicated that this was due to patients expressing a desire for low doses, but there is also a need to justify this clinically. Methods:  - Page 4 Line 42 "We also received feedback from a group of TBI patients through a Clinical Psychologist who acted as a facilitator" – It is unclear who this group of TBI patients were (participants, other patients etc.) or when/why/how they were consulted for feedback.
---

- Page 4 line 51 –Please describe the centres that participants came from in greater detail. It later says that they are neuropsychiatry clinics but more information is required here.
- The visual appearance of placebo capsules is not described, meaning that it is difficult to tell whether this study is truly blinded. It is also unclear as to exactly who was blinded to study treatment, beyond the clinician making the decision to escalate dosage.
- There is no description of adherence monitoring. It is therefore unclear if the participants actually took the study product and if the results can be trusted.
- It needs to be explicitly stated what each outcome measure is measuring, and provide information regarding their reliability and validity. This is done for the main outcome measure (MAOS) but not the secondary measures.
- There are many discrepancies between the protocol and the final publication. For example, some measures described in the protocol aren't included in the final report (e.g. CSRI), bloods results etc. Inclusion and exclusion criteria are also different. Authors should include a detailed deviation from protocol table with justifications for each deviation.
- Were there any measures used to operationalise presence of 'aggression' to allow entry in to the study? Or was it just clinician judgement?
- PG 6 LN 10 – what definition was used for 'major clinical improvement' and 'troublesome adverse events' that necessitated change in dose?
- PG 8 LN 33 could the sentence 'the interviews were conducted using an interview topic' be expanded? It is not clear what the topic was or whether there was one topic per interview?
- PG 8 LN 42 it would be of interest to include a list of topics covered in the final iteration of the interview
- PG 8 LN 15 it is not clear what is meant by the study received a 'favourable ethical opinion'?

Results:

- Table 1:
 - o How was TBI severity defined?
 - o "history of amnesia" is somewhat misleading – are they referring to PTA?
- The process evaluation was very insightful
- Given there were only 7 people included in the trial, I think it would be interesting (and feasible) to include more detail regarding the dose of risperidone each person received. PG 6 LN 1 – 15 states that the dose was initially the same but then was titrated with clinical need, along with PRN dose being acceptable also.
- The variability on the MAOS is much greater for the placebo group – can the authors comment on this and whether it is due to outliers and how this may have influenced the group differences.

Discussion:

- The authors barely discuss the results of the aggression measures, any implications of this or the limitations associated with the kind of analysis they were able to do.

General Notes:

- Many grammatical revisions are required. For example, "They advised us to be mindful of adverse effects of risperidone preferred a smaller dose for the RCT." (Pg 4, Line 31), "we have incorporated within the design a through examination..." (Pg 4, Line 36). Clinical Psychologist should not be capitalised (Pg 4 line 42). This was just one page. Many more examples throughout.
- The paper needs improvement in flow and structure, especially in the introduction and discussion.

	- Abbreviations are used without being introduced in full (e.g. CSO, page 12, line 24).
--	---

VERSION 1 – AUTHOR RESPONSE

Reviewer: 1

Reviewer Name Malcolm Hopwood Institution and Country Department of Psychiatry University of Melbourne Australia Please state any competing interests or state 'None declared':

None declared Please leave your comments for the authors below

This submission describes a well-structured and well described RCT of risperidone versus placebo for post ABI aggression. This type of study is clearly needed in this relatively low evidenced area of practice and where the balance of harms and benefits needs careful consideration. The background to the study contains much of the relevant literature. The study design is well laid out and not really open to much criticism. There are several instruments available for the measurement of aggression in this setting and the choice of the MOAS, whilst not unreasonable, could be discussed a little. The dose range utilised in the study is consistent with common clinical practice although it may be worth considering and discussing whether an even lower dose such as 0.5 mg was worth initial trial given the sensitivity issues in this group. The results of the study are laid out well with the main problem in their interpretation being the obvious difficulties with recruitment. The authors are appropriately statistically conservative given the low numbers but do draw some tentative conclusions about the relative efficacy of placebo and risperidone. Interestingly, the results in the risperidone arm were actually numerically better when looking at carer ratings of aggression, and this result should be considered a little more in my view given that we know there is a high level of disparity between patient and carer ratings of aggression in this setting. The major focus of the paper then is on the struggle with recruitment and the discussion around this area is of value as this is a problem that has dogged many studies in this specific area. This may be the most valuable contribution of the paper and is thus definitely worth retaining.

We have now justified our choice of the MOAS (see Table 1 and 2).

We have also discussed in detail the issue of risperidone dosing both in the Introduction section and in the Discussion section.

We have now addressed the discrepancy in patient vs. carer reported MOAS values in the Discussion section.

Reviewer: 2

Reviewer Name Durga Roy, MD Institution and Country Johns Hopkins University School of Medicine Department of Psychiatry and Behavioral Sciences

Please state any competing interests or state 'None declared': None Please leave your comments for the authors below

While the study question and intervention is interesting, the manuscript is poorly written with a substantial number of grammatical and syntax errors, cursory literature review on TBI aggression and unclear rationale described in reference to parts of the study protocol. The methods appear to be sound, but are not organized with appropriate syllogism to guide the reader as to the rationale for many components of the study design. The most overarching concern is that while this is a feasibility study, the authors use this as a reason for not having completing a power analysis or even establishing a hypothesis. Also there is no discussion of adjusting for confounding factors such as psychiatric comorbidities (not symptoms , but syndromes) that exist with TBI aggression. For a data driven paper, in the very least the reader would most certainly benefit from hearing about what the

authors hypothesize would be the outcome of using an intervention like Risperidone to manage aggression. I have listed my comments and questions based on each section:

Introduction

- The background literature in the introduction is extremely underdeveloped. Some of the most landmark studies in TBI aggression (Tateno et al, etc) are not cited. Would suggest a more robust lit review on this topic.

As ours is a feasibility study we did not put much emphasis on post-TBI aggression and its treatment in the Introduction section and the Discussion section in our previous submission. However, as asked by the reviewer, we have now presented a robust, comprehensive up to date literature review on post-TBI aggression and its pharmacological intervention.

- In the first paragraph would clarify what is meant by “medical advances” line 45 page 3.

This line does no longer appear in the completely re-written Introduction section in the revision.

- On page 4 line 3, the authors mention that aggression is the most challenging post-TBI neurobehavioral symptom but what is the prevalence?

See Table 1.

- Aggression is not defined anywhere in the intro. It should also be made very clear that aggression is not the same as agitation and though neuroleptics have been used in the management of both it needs to be made clear that this paper will focus on aggression only.

We have now defined ‘aggression’ in the Methods section and clarified that ‘agitation’ is excluded.

- Line 10 the authors mention “Both pharmacological and non-pharmacological interventions are used to manage this difficult behavior” but there is no reference cited following this sentence to substantiate the claim. This is a key piece of background information as the paper hinges on a pharmacologic intervention to manage TBI aggression.

We have now presented a comprehensive literature review on the pharmacological interventions of post-TBI aggression (see Table 2).

- Though it is clear that there is no high quality evidence of trials, namely they were not RCT, what were the limitations the findings in the literature ? were they only case repots or very underpowered open label studies?

See Table 2.

- Would explain to the reader why second generation antipsychotics are used to manage aggression in this patient population. – namely the side effect profile

Now presented in the revised Introduction section.

- On page 5 line 37- why is it that clinicians tend to choose antipsychotics? Would talk about the literature behind these practices.

Now presented a comprehensive literature review on the subject in the Introduction section.

- Page 5 Line 40 is a run on sentence and needs to be revised,

Page 5 line 40 in the previous version: “patient suffering from Post-Traumatic Amnesia (PTA)” is one of the exclusion criteria.

- The rationale behind why the authors chose risperidone as the intervention for this study is not clear. Just because it has been used in the other neurobehavioral syndromes does not necessarily make it an obvious intervention in TBI. It is well documented that higher potency neuroleptics can impair neuronal growth. The authors should delve more into the pharmacokinetic, pharmacodynamic, efficacy and effectiveness reasons that this drug was chosen.

This issue including pharmacokinetics, pharmacodynamics, efficacy and effectiveness is now addressed in detail both in the Introduction and the Discussion section.

Methods

- In the methods section, subjects were recruited from a Headway center. Is this a residential center? Might be helpful if the setting is clear to the reader whether this is an inpatient, outpatient, rehab or residential setting.

Participants were not recruited from any Headway Centre. We have now clarified that participants were recruited from three neuropsychiatric outpatient clinics and one neurology clinic dedicated to TBI patients.

- Line 31 on page 6 does not make grammatical sense

Page 6, line 31 in the previous version: “Primary outcome measure.”

- What is a trials management group and what did it mean that the subject was a co-applicant.

We have now changed this to ‘project group’ and removed the sentence ‘subject was a co-applicant.’

- What was the study feedback from participants done through focus groups? (page 4 line 42)

We have summarised the focus group findings in the Results section. However, given the space available it is not possible to present all focus group data here. In this paper we could only present summary findings of both feasibility/ clinical outcome data and focus group discussions.

- I am not clear as to why there was no power calculation. The authors cite the reason as this being a feasibility study, however this should not preclude to the study design to include a power analysis.

A power calculation is needed for a ‘pilot study’ but not for a ‘feasibility study.’ In fact, often feasibility studies are done to collect data for calculation of sample size for a future definitive RCT, which is the intention in our study. The aim of our study was not to show a statistical difference between the groups, but to focus on the practical aspects of running the trial. i.e. on the ability to recruit, collect data etc. See the following references;

Lancaster GA, Dodd S, Williamson PR. Design and analysis of pilot studies: recommendations for good practice. *Journal of Evaluation in Clinical Practice* 2002;10:307-12.
Browne RH. On the use of a pilot sample for size determination. *Statistics & Medicine* 1995;14:1933-40.

Field A (2005). *Discovering Statistics using SPSS* (2nd Ed.). London: Sage.

- Why were patients with PTA excluded? If there was a need to exclude these patients then what about those that had LOC or AMS? A rationale as to why those with PT were excluded is something that needs to be clarified.

Patients with a history of PTA were not excluded (see Table 3) but were not included if they were in PTA at the time of recruitment, which of course is unlikely at 6 months post-TBI.

- How was dosing of Risperdal determined? It is reported by authors that dosing was “flexible” but

We have now discussed the dosing issue in detail in the Introduction section and the Discussion section.

- Similar to the above comment on no power analysis- in the statistical analysis why was there no hypothesis tested? Page 7 line 36. Even as a feasibility study, it would be useful to the reader if it could be determined what outcomes the authors would expect from a study like this. Which would also require a more thorough review of the literature about what has been done in pharmacological management of TBI aggression.

To be sufficiently powered to show a minimally clinically important difference between groups, a much larger study would have been needed. Hence, with the sample size in the order of magnitude of this study, the study would have had a very low power to show a difference. In other words, very much underpowered. The data could have shown a difference in outcome between groups, but the sample size would not have been sufficient to show this statistically. As there was no hypothesis testing, there was no power calculation. Such a calculation is only appropriate if it relates to a statistical test.

- On page 8 line 34, what is meant by interviews were conducted by interview topic? This is very ambiguous and requires much more detail. It is mentioned above that a semi-structured

We have now summarised the broad headings of the interview topic guide in the Methods section. There is no space to provide full findings from the process evaluation work, which will require another publication. Given the space, we have only summarised the main quantitative and qualitative findings and feasibility outcome.

Results

- Why were 80% of patients on risperidone (should not read risperidone patients) mildly or less ill at the 12 week timepoint? Is it because they had a full trial duration of the neuroleptic? Also, what is meant by “mildly ill?” Are these in reference to psychiatric symptoms?

These are standard CGI-I categories and not related to psychiatric symptoms as no participant with psychiatric diagnosis was included as the presence of serious psychiatric disorders such as schizophrenia was an exclusion criterion.

- How was aggression due to psychiatric comorbidity adjusted for? It is well established that aggression can co-occur with depression, mania, anxiety and psychosis in TBI, so how was this incorporated into the analysis.

Psychiatric diagnoses such as schizophrenia, major depressive disorder, bipolar disorder, personality disorder and dementia were exclusion criteria, so no participants had any of these psychiatric disorders.

- Was the placebo group on any other psychotropic? I am not clear about how the intervention was randomized to each group if the goal was to see if aggression was reduced in the risperidone group. There should also be mention of IRB approval somewhere in the text for the study protocol.

We have now mentioned that no participant received any concomitant psychotropic medication including PRN. This would be expected as we excluded participants with psychiatric disorders. In the UK, there is no provision for IRB approval. However, we have mentioned that the study was approved by the Local Ethics committee.

Discussion

- This section is quite superficial and doesn't delve into the implications of the study findings. What can the reader/clinician take away from these outcomes? Understood that it is a feasibility study but some could some preliminary conclusions be drawn?

We have now expanded the discussion on the clinical outcome in the Discussion section, space permitting.

- There are no limitations listed in the discussion and this is a standard component of this section

We have now added a 'strength and weakness' section at the end of the Discussion section.

Overall this paper could be of some value in the first steps of learning about management of TBI aggression but the lack of solid scientific rigor in the study design and statistical analyses, very cursory review of the literature on TBI aggression and underdeveloped syllogistic writing style are detrimental to the final product.

Reviewer: 3

Reviewer Name Professor Jennie Ponsford Institution and Country Monash University, Australia
Please state any competing interests or state 'None declared': None declared
Please leave your comments for the authors below
This study promised to fill a significant gap in evidence for pharmacological management of aggression after traumatic brain injury. Unfortunately the study met with many hurdles regarding feasibility. The following comments are intended to improve the manuscript.

Abstract:

- Page 2, line 5: It is misleading to say that overall rate of adverse events was non-significantly higher in the risperidone group, as no significance testing was conducted. This might also lead the reader of the abstract to believe that other findings were hypothesis tested, which they weren't. It should also be stated in the abstract that two people withdrew from the study due to adverse events.

Agreed. Now removed. We have also added that two people withdrew from the study due to adverse events.

- Page 2, Line 12: More detail could be added regarding what measures may increase recruitment, even if only a couple. This is very vague at present.

We have now provided a list under the heading 'recommendation from the feasibility study' within the Discussion section. We have now added a few recommendations in the Abstract, space permitting.

- Page 12, 36: It is stated that many regulatory authority delays hampered patient recruitment in the abstract but don't discuss this again. This should surely be added to the discussion if it is significant enough to appear in the abstract.

We have now added this in the Discussion section. However, as the regulatory process varies from country to country, we did not go into any detail about this. As we had to condense the text on feasibility outcome in the Discussion section to save space, we have included this issue within the 'recommendation from the feasibility study' section.

Introduction:

- The published protocol introduction has been copied almost exactly. Published protocol link: <https://trialsjournal.biomedcentral.com/articles/10.1186/s13063-018-2601-z>

The Introduction section is now completely re-written.

- P3 Line 16 to 20 – there are existing published studies examining medication prescribing for neurobehavioural problems post ABI – it would be best to cite one of these rather than unpublished data
- P3 Line 23 – A more recent systematic review confirming the lack of high-quality clinical trials in this area is : Hicks, A.J., Clay, F.J., Hopwood, M., James, A., Jayaram, M., Perry, L.A., Batty, R. & Ponsford, J.L. (2019). The Efficacy and Harms of Pharmacological Interventions for Aggression after Traumatic Brain Injury – Systematic Review. *Frontiers in Neurology*. Published online 29 November 2019. DOI: 10.3389/fneur.2019.01169.

We have now presented a comprehensive and up to date literature review on this in the Introduction section. We have included data from Hicks et al. (2019) paper but excluded Brossart et al. (2008) paper as it looked into agitation rather than aggression and as the data were presented within the context of a detailed description of a specific statistical methodology, it is very difficult to interpret the findings.

- P3 Line 32 – Please justify why second generation antipsychotics are chosen more frequently than first generation compounds.

We have now discussed this in detail in the Introduction section.

- P3 Line 39 –Please include a reference for the claim that clinicians use their 'often limited experience' to choose antipsychotics to treat aggression post TBI - P3 Line 45 (and elsewhere)

We have now presented data on the evidence available for the pharmacological intervention of post-TBI aggression in the Introduction section showing that good quality evidence is lacking to guide the clinicians on the ground.

– Further explanation is required to justify the decision to administer low dose risperidone. It was indicated that this was due to patients expressing a desire for low doses, but there is also a need to justify this clinically.

We have discussed this now in detail both in the Introduction and the Discussion section.

Methods:

- Page 4 Line 42 “We also received feedback from a group of TBI patients through a Clinical Psychologist who acted as a facilitator” – It is unclear who this group of TBI patients were (participants, other patients etc.) or when/why/how they were consulted for feedback.

We have now clarified in the Methods sections that these TBI patients who provided advice to the project group did not take part in the study. We also clarified that these TBI patients attended a day centre which was not associated with any of the recruiting centres and met every few months to provide feedback to the study.

- Page 4 line 51 –Please describe the centres that participants came from in greater detail. It later says that they are neuropsychiatry clinics but more information is required here.

We have now clarified this in the Methods section. We clarified that the participants were recruited from outpatient clinics (three neuropsychiatry and one neurology clinic dedicated to TBI patients).

- The visual appearance of placebo capsules is not described, meaning that it is difficult to tell whether this study is truly blinded. It is also unclear as to exactly who was blinded to study treatment, beyond the clinician making the decision to escalate dosage.

We have now clarified in the Methods section that the placebo capsules looked exactly the same in shape, colour, contour as the IMP. We have also clarified that everyone involved in the study including patients, their carers, the researcher, participant’s general practitioners, treating clinicians were blind to the IMP allocation.

- There is no description of adherence monitoring. It is therefore unclear if the participants actually took the study product and if the results can be trusted.

We have now added in the Methods section under ‘active drug dose’ section how we assessed adherence and reported in the Discussion section under ‘Risperidone dose’ section that there was no evidence of lack of adherence.

- It needs to be explicitly stated what each outcome measure is measuring, and provide information regarding their reliability and validity. This is done for the main outcome measure (MAOS) but not the secondary measures.

We have now expanded this section in the Methods section under ‘Outcome measures’ heading as much as possible given the limited space.

- There are many discrepancies between the protocol and the final publication. For example, some measures described in the protocol aren’t included in the final report (e.g. CSRI), bloods results etc. Inclusion and exclusion criteria are also different. Authors should include a detailed deviation from protocol table with justifications for each deviation.

We have collected data on CSW and CSRI but because of lack of space and small sample size we did not present these data in the paper. As we did not present these data, we have not mentioned about these measures in the Methods section as it will confuse readers. Also, if we mention these in the Methods section and then say why we have not presented the data, this will take up much space unnecessarily and also be superfluous.

We have mentioned about collecting blood test results where available in the participant’s case notes in the Methods section. We have also mentioned in the Discussion section that no relevant post-intervention blood tests were available from participants’ case notes.

In our protocol paper in Trials we had more space to go into the detail of all the inclusion exclusion criteria. However, because of lack of space in the current paper where we wanted to present only summary findings, we have chosen to report only the important inclusion exclusion criteria in the current paper. We have also mentioned now in the beginning of the Methods section that for a fuller description of the design the readers should read our protocol paper in Trials. For example, in the current paper we included “any other contraindication for using risperidone” within exclusion criteria but in the protocol paper we have provided much more detail about these contraindications.

Although we don't think it is necessary but if the reviewer insists and the editor is happy for us to increase the length of the paper, which is already over the word limit after complying with all reviewers' comments, we will be happy to report the full inclusion exclusion criteria.

There is no deviation from protocol to report.

- Were there any measures used to operationalise presence of 'aggression' to allow entry in to the study? Or was it just clinician judgement?

We have now mentioned in the Methods section how we defined aggression and clarified that no arbitrary cut off score measure was used to operationalise 'aggression' to allow entry into the study.'

- PG 6 LN 10 – what definition was used for 'major clinical improvement' and 'troublesome adverse events' that necessitated change in dose?

We have now clarified this in the Methods section.

- PG 8 LN 33 could the sentence 'the interviews were conducted using an interview topic' be expanded? It is not clear what the topic was or whether there was one topic per interview?

We have now provided the broad headings of the topic guide in the Methods section. However, as process evaluation has produced a rich dataset, it is not possible to present all the detail on this in this paper, which will be presented in a separate paper. In this paper we only have space to summarise the main findings.

- PG 8 LN 42 it would be of interest to include a list of topics covered in the final iteration of the interview

See above.

- PG 8 LN 15 it is not clear what is meant by the study received a 'favourable ethical opinion'?

In England Ethics committees have now changed the wording and instead of stating that 'the study has been approved' they state that 'a favourable ethical opinion is given' which is the same as stating that the study received an ethics approval. For those who are not familiar with these wordings, we have now changed the sentence to 'The study received ethical approval.' to avoid any confusion.

Results: - Table 1:

o How was TBI severity defined?

• “history of amnesia” is somewhat misleading – are they referring to PTA?

Mayo Clinic criteria was used to define TBI severity (see Methods section).

It included all types of amnesia but for all practical purposes, it is primarily PTA. We have now changed the wording from 'amnesia' to 'PTA.'

- The process evaluation was very insightful

This is at the heart of the feasibility study. We could only summarise the main points here given the lack of space.

- Given there were only 7 people included in the trial, I think it would be interesting (and feasible) to include more detail regarding the dose of risperidone each person received. PG 6 LN 1 – 15 states that the dose was initially the same but then was titrated with clinical need, along with PRN dose being acceptable also.

We have now provided data on risperidone dose in the Results section and discussed this further in the Discussion section. We also mentioned that no one needed PRN medication, and no one was on any other concomitant psychotropic medication.

- The variability on the MAOS is much greater for the placebo group – can the authors comment on this and whether it is due to outliers and how this may have influenced the group differences.

This is not due to an outlier; it is more that the MOAS score has a very positively skewed distribution. Some participants do have very large scores, but this is typical of the distribution. As a result of the skewed distribution, the median/ inter-quartile range are probably the most appropriate summary measures. The mean/ SD were also reported primarily as these could be useful to somebody who might require the information to power a further study in the future.

Discussion:

- The authors barely discuss the results of the aggression measures, any implications of this or the limitations associated with the kind of analysis they were able to do.

We did not do this in the previous submission because of the very small sample size which may give a false interpretation of outcome to the readers and may be seen as definitive. However, we have now devoted more space discussing clinical outcome in the Discussion section as much as is allowed within the restricted space available.

General Notes:

- Many grammatical revisions are required. For example, "They advised us to be mindful of adverse effects of risperidone preferred a smaller dose for the RCT." (Pg 4, Line 31), "we have incorporated within the design a through examination..." (Pg 4, Line 36). Clinical Psychologist should not be capitalised (Pg 4 line 42). This was just one page. Many more examples throughout.
- The paper needs improvement in flow and structure, especially in the introduction and discussion.

We apologise for this. This happened when all the tracker changes used by co-authors were amalgamated and possibly inadvertently an unedited version was uploaded.

- Abbreviations are used without being introduced in full (e.g. CSO, page 12, line 24).

Done.